# Prevalence of chronic pulmonary aspergillosis along the continuum of pulmonary tuberculosis care: A protocol for a living systematic review and meta-analysis

Felix Bongomin[1,2]*, Ronald Olum[3], Winnie Kibone[2], Martha Namusobya[4], Norman van Rhijn[1], David W. Denning[1]*

1 Manchester Fungal Infection Group, Division of Evolution, Infection and Genomics, School of Biological Sciences, Faculty of Biology, Medicine and Health, University of Manchester, Manchester, United Kingdom, 2 Department of Medical Microbiology and Immunology, Faculty of Medicine, Gulu University, Gulu, Uganda, 3 School of Public Health, College of Health Sciences, Makerere University, Kampala, Uganda, 4 Department of Epidemiology and Biostatistics, College of Health Sciences, Makerere University, Kampala, Uganda

* felix.bongomin@postgrad.manchester.ac.uk (FB); david.denning@manchester.ac.uk (DWD)

**Data Availability Statement:** No datasets were generated or analysed during the current study. All relevant deidentified research data from this study

## Abstract

### Introduction

Chronic pulmonary aspergillosis (CPA) is a debilitating disease estimated to affect over 3 million people worldwide. Pulmonary tuberculosis (PTB) is the most significant risk factor for CPA. However, the true burden of CPA at the time of PTB diagnosis, during, and after PTB treatment remains unknown. In this paper, we present a protocol for a living systematic review aimed at estimating the current burden of CPA along the continuum of PTB care.

### Materials and methods

We followed the Preferred Reporting Items for Systematic Reviews and Meta-Analyses Protocols (PRISMA-P) guidelines to formulate this protocol, which is registered with the International Prospective Register of Systematic Reviews (PROSPERO: CRD42023453900). We will identify primary literature through various electronic databases, including CINAHL, Ovid MEDLINE, MEDLINE (PubMed), EMBASE, Google Scholar, Cochrane Database of Systematic Reviews, and African Journal Online. The search will encompass articles from inception to December 31st, 2023, using medical subject heading search terms "pulmonary tuberculosis" AND "chronic pulmonary aspergillosis". Two reviewers will independently assess titles, abstracts, and full texts for eligibility using the Covidence web-based software. The eligible studies will comprise original observational research that reports on the prevalence of CPA diagnosed in individuals with PTB, based on established criteria, without language or geographic restriction. We intend to exclude single case reports and case series with fewer than 10 participants, as well as review articles, guidelines, and letters to the editors. Cochrane Risk of Bias Tools (ROB2 and ROBINS-I) will used to assess study quality and risk of bias and the quality of the evidence will be rated using the Grading of

will be made available upon study completion and publication.

**Funding:** This work was financially supported by the Carigest SA Conny Naeva Charitable Foundation as part of PhD studentship award to Dr Felix Bongomin at the University of Manchester, United Kingdom. No additional external funding was received for this study. The funder had no role in study design, data collection and analysis, decision to publish, or preparation of the manuscript.

**Competing interests:** The authors have declared that no competing interests exist.

Recommendations, Assessment, Development and Evaluations (GRADE) tool. Our data syntheses will encompass meta-analysis and meta-regression, conducted using STATA version 18 and R- Studio version 4.0.2. This systematic review will be updated every 3–5 years as more data emerges.

## Conclusions

The findings of this proposed systematic review will summarize the available evidence on the occurrence of CPA, at the time of PTB diagnosis, during and after PTB treatment. The study results have the potential to guide healthcare policies regarding screening for CPA, enhance clinical decision-making, and catalyse further research into understanding the interplay between PTB and CPA. By shedding light on the current burden of CPA along the continuum of PTB care, we aspire to contribute to the betterment of patient care, disease management, and global health outcomes.

## PROSPERO registration

CRD42023453900.

## Introduction

Chronic pulmonary aspergillosis (CPA) is a slowly progressive and destructive parenchymal lung disease mostly caused by *Aspergillus fumigatus* and affecting both immunocompetent and subtly immunocompromised patients, particularly those with previous or underlying structural lung damage due to infectious and non-infectious diseases such as pulmonary tuberculosis (PTB), fibrocystic sarcoidosis, non-tuberculous *Mycobacterium*–pulmonary disease, and others [1, 2]. Of these, active, or previous treated PTB is the most common with prevalence ranging between 15% and 90% of patients with CPA [3, 4].

A major multi-center, prospective cohort study in the mid- to late-1960s across clinics in Great Britain confirmed that residual cavities following treatment of PTB plays a crucial role in the development of CPA [5]. In this study, 14% of participants had aspergilloma after one year, and this number increased to 22% after three years of follow-up. Several other studies later on confirm the hypothesis that CPA complicates residual cavitation following PTB treatment [6–13].

The global burden of PTB-associated CPA is unknown. However, a deterministic modelling of the global burden of CPA estimated that about three million people worldwide. Of these, approximately 1.2 million cases are thought to be due to previously treated PTB. Recent estimates from Africa has shown that PTB accounts for over 60% of underlying disease among over 1,200 published cases of CPA across the continent [14–16]. A recent study in India estimated an annual incidence of 363,601 cases of CPA associated with PTB and 42,766 deaths, accounting for 11% of total PTB deaths. Once later occurrence of CPA is factored in, a 5-year-period prevalence of 1,575,716 cases was estimated with a total of 142,484 deaths (including the incident case mortality). This highlights the significant impact of CPA as a complication of PTB, emphasizing the need for substantial investment in its diagnosis and integration into PTB care, both in the short and long term [17]. This point is further emphasized by the key drug-drug interaction between rifampicin and antifungal azoles which requires differentiation of the entities and not simply treating both together as antifungal drug exposure falls to zero if given with rifampicin.

Our current understanding of the spectrum of CPA in a population of PTB patients has broadened, with observational studies showing that CPA occurs in about 8% of patients being treated as smear-negative or PTB relapse [7], 4 to 20% in early PTB or end of PTB treatment and in patients with persistent symptoms despite optimal anti-tuberculous treatment [8, 10, 14, 18], and 1 to 54% post-PTB treatment [6, 8, 9, 11, 13, 14, 19].

Much as CPA was previously thought to be predominantly occur in patients with previously treated PTB, empirical evidence supports the occurrence of CPA at diagnosis (as a misdiagnosis), during and at the end of PTB treatment (as a co-infection) and in those who have completed PTB treatment (as a complication). Therefore, this systematic review and meta-analysis aims at comprehensively pooling data on the prevalence of CPA across the continuum of PTB care to inform screening, management and integration of CPA in TB programs and clinics.

## Broad objective

To determine *Aspergillus* antibody seropositivity rate, species of isolates of *Aspergillus* and the prevalence of CPA among patients with active or treated PTB.

## Specific objectives

1. To determine the pooled prevalence of CPA among people with active PTB

2. To determine the pooled prevalence of CPA among people with successfully treated PTB.

3. To determine the pooled prevalence of CPA among patients with smear-negative or radiologically diagnosed PTB.

4. To evaluate the pooled proportion of people with active PTB or treated PTB with elevated *Aspergillus* antibody or positive *Aspergillus* precipitins.

5. To identify species of *Aspergillus* isolated in patients with active or treated PTB.

# Methods and materials

## Protocol and registration

This systematic review and meta-analysis will be conducted following the Preferred Reporting Items for Systematic Reviews and Meta-Analyses (PRISMA) guidelines [20], including the PRISMA-Protocol (PRISAM-P) checklist. The protocol has been registered with the International Prospective Register of Systematic Reviews (PROSPERO: CRD42023453900).

## Information sources

A comprehensive literature search will be conducted in all the relevant scientific databases and grey literature to identify all articles related to pulmonary tuberculosis and aspergillosis. We will search MEDLINE, EMBASE, CINAHL, and Global Health through the OVID interface from 1946 to date with no language and geographical limitations. We will also search the Cochrane Central Registry of Controlled Trials and the Cochrane Database of Systematic Reviews. To identify relevant articles not indexed in the databases above, we will conduct a keyword search on Google Scholar and export the first 500 references sorted by relevance. To minimize publication bias, we will explore grey literature sources such as abstracts of major conferences on tuberculosis and/or mycoses, including annual scientific meetings by international and national mycological societies or interest groups. In addition, forward and

**Table 1. Search strategy.**

| Concepts | Search Terms |
|---|---|
| Pulmonary tuberculosis | "Tuberculosis, pulmonary" [MeSH Terms] OR "pulmonary tuberculosis" OR "active tuberculosis" OR "active TB" OR "active PTB" OR "PTB" OR "lung TB" OR "tuberculous pneumon\*" OR "mycobacterium tuberculosis" OR "mycobacter\* tuberculosis" OR "phthisis" OR "phthisis pulmonalis" OR "consumption" OR "Koch's disease" |
| Pulmonary aspergillosis | "pulmonary aspergillosis"[MeSH Terms] OR "pulmonary aspergillosis" OR "aspergillus lung infection" OR "lung aspergillosis" OR "aspergilloma" OR "mycetoma of the lung" OR "pulmonary mycetoma" OR "chronic pulmonary aspergillosis" OR "cpa" OR "chronic necrotizing pulmonary aspergillosis" OR "cnpa" OR "invasive pulmonary aspergillosis" OR "ipa" OR "acute invasive aspergillosis" OR "allergic bronchopulmonary aspergillosis" OR "abpa" OR "simple aspergilloma" OR "complex aspergilloma" OR "semi-invasive aspergillosis" |

backward snowballing of the references of the included articles will be performed to find relevant studies.

## Search strategy

The search terms for this systematic review and meta-analysis have been developed after extensive consultation with experts and librarians. First, two main concepts were formed–pulmonary tuberculosis, and chronic pulmonary aspergillosis. The search strategy was then built by including all the possible terminologies, synonyms and alternative words used to describe the two concepts. The strategy was first tested and refined in MEDLINE using medical subheadings (MESH), keywords and free texts, and thereafter adopted for the rest of the databases. The MEDLINE search strategy is included in Table 1 below.

## Data management

Results from the literature search will be exported and uploaded to EndNote Library for merging and deduplication. The final references will then be uploaded to Covidence, an internet-based software, for screening and extraction. The details of the number of articles at each level of the systematic review and meta-analysis will be recorded into the PRISMA Flow Diagram.

## Eligibility criteria

The inclusion and exclusion criteria for this systematic review has been developed using the *P*opulation, *E*xposure, and *O*utcome (PEO) framework. We will include all studies involving human participants regardless of the age, gender, ethnicity, or geographic location, reporting the prevalence or incidence of any form of CPA in individuals with a history microbiologically proven and or clinically diagnosed active or treated PTB. Both observational (cross-sectional, case-control, cohort) and interventional studies (randomized controlled trials, quasi-experimental studies) that provide data on the prevalence or incidence of pulmonary aspergillosis among individuals with pulmonary tuberculosis will be included. Systematic reviews and meta-analyses that offer additional primary studies not captured in the initial search will also be included. No language or date limitations will be applied to the initial review. For the subsequent reviews, the search shall be limited to the last date of the previous review and the current date of literature search. Editorials, opinion pieces, case reports, case series with fewer than 10 participants, and primary studies with unclear methodology or insufficient data will be excluded.

## Screening process

A minimum of independent reviewers will independently screen all articles by title and abstracts using Covidence Software to determine if they meet the eligibility criteria. Full texts of all articles deemed eligible at this phase will then be retrieved, reviewed, and assessed for eligibility by at least two independent reviewers. Where full texts of the articles cannot be retrieved, the primary authors will be contacted for additional information. Any disagreements during the screening process will be resolve through discussion and consensus. The reasons for exclusion shall be recorded all articles at the full-text screening.

## Data extraction

A standardized form with a detailed manual on the variables to be extracted will be developed in Microsoft Excel and pre-tested prior to data extraction. Data from each full-text article will be extracted and entered in duplicate by at least two independent reviewers for consistency and quality assurance. All disagreements will be resolved by discussion and consensus between the authors, and the principal investigators. In case of any uncertainties about the study methods or results, an attempt to contact the primary authors will be made to seek clarity.

## Study variables

The following variables will be extracted from each included article:

1. Study Information

   a. Study Identification: Author(s), publication year, journal title.

   b. Study Design: Type of study (e.g., cross-sectional, cohort, case-control, randomized controlled trial).

   c. Study Setting: Geographic location, study site (e.g., hospital, community), and study period.

   d. Sample Size: Total number of participants and, if relevant, number in each study arm or group.

2. Participant Information

   a. Demographics: Age, gender, ethnicity, and other relevant sociodemographic information.

   b. Clinical Characteristics: Comorbidities, history of previous TB episodes, HIV status, and any other pertinent clinical data.

3. Exposure Information

   a. TB Categories such as active TB, treated TB, smear-negative TB, and radiologically diagnosed TB.

   b. Duration of TB: Length of time since TB diagnosis, treatment duration (if applicable).

4. Outcome Variables

   a. CPA Diagnosis: Methods used for diagnosis (e.g., chest imaging findings, fungal culture, serological tests).

   b. Type of CPA: E.g., chronic cavitary pulmonary aspergillosis, simple aspergilloma, chronic fibrosing pulmonary aspergillosis, *Aspergillus* nodules or subacute invasive pulmonary aspergillosis.

 c. *Aspergillus* species: Specific species identified, e.g., *A. fumigatus*, *A. niger* etc, if available.

 d. *Aspergillus* antibody Seropositivity Rate: Including specific tests used, if mentioned.

5. Results and Findings

 a. Key Outcomes: Prevalence rates, odds ratios, hazard ratios, or any other relevant statistics reported.

6. Methodological Quality and Bias Indicators

 a. Sampling Method: E.g., random sampling, convenience sampling.

 b. Response Rate: Percentage of participants who completed the study or follow-up.

 c. Confounding Variables: Any variables that were adjusted for in analyses.

 d. Limitations: As acknowledged by the study authors.

## Study outcome

a. Primary outcome

- Prevalence of CPA defined as an illness for >3 months and all of the following: 1) weight loss, persistent cough, and/or haemoptysis; 2) chest images showing progressive cavitary infiltrates and/or a fungal ball and/or pericavitary fibrosis or infiltrates or pleural thickening; and 3) a positive *Aspergillus* IgG assay result or other evidence of *Aspergillus* infection [1, 21].

b. Secondary outcomes

- Prevalence of isolation of *Aspergillus*
- Prevalence of positive *Aspergillus* serology
- *Aspergillus* antifungal resistance profile
- Mortality/deaths
- Recurrence
- Treatment success
- Clinical presentation
- co-morbidities
- Underlying risk factors.

## Risk of bias assessment

To ensure the validity and reliability of the findings from this systematic review and meta-analysis, a rigorous assessment of the risk of bias in the included studies will be conducted. For randomized controlled trials, the Cochrane Risk of Bias (ROB-2) tool will be used [22]. For non-randomized studies, the Risk of Bias in Non-randomized Studies—of Interventions (ROBINS-I) tool will be employed. Both tools are designed to provide a detailed evaluation of potential biases across multiple domains [23]. Two independent reviewers will undertake the

risk of bias assessment for each study. Disagreements between the reviewers will be resolved through discussion or, if needed, consultation with a third reviewer. Each domain will be rated as 'low risk', 'unclear risk', or 'high risk' of bias, based on the criteria outlined in the respective tools. For observational studies, the modified Newcastle Ottawa Scale will be used to assess the risk of bias [24].

The risk of bias assessments will be graphically represented in the final report using a 'Risk of Bias' graph, offering a visual summary of the overall quality of the included studies. Additionally, a narrative synthesis will detail the main sources of bias identified across studies. Should there be a considerable number of studies with high or unclear risks of bias, sensitivity analyses will be conducted to determine the potential impact of these studies on the overall meta-analysis results.

## Data synthesis

An initial narrative synthesis will be used provide a description of the characteristics and findings of all included studies, detailing their design, population, exposure, outcomes, and key results. If appropriate and feasible, a meta-analysis will be conducted to quantitatively synthesize the findings of studies reporting comparable outcomes. Using a random-effects model, we will calculate the pooled prevalence of chronic pulmonary aspergillosis in various PTB populations. This model is preferred given the anticipated heterogeneity among studies. The $I^2$ statistic will be employed to quantify the proportion of total variation in study estimates due to heterogeneity. An $I^2$ value > 50% will indicate substantial heterogeneity.

If sufficient data is available and appropriate, subgroup analysis shall be conducted to explore the potential sources of heterogeneity. The subgroup analyses will be based on geographical region or settings, study design, subtypes of CPA, population demographics, and diagnostic criteria among others. Should significant heterogeneity be identified among included studies, a multilevel meta-regression will be conducted to explore potential sources of this variability and better understand the influence of specific study-level covariates on the pooled prevalence of chronic pulmonary aspergillosis.

To assess the robustness of our findings, sensitivity analyses will be conducted by excluding studies with a high risk of bias and outliers that might overly influence the pooled estimates. The potential for publication bias will be evaluated using funnel plots, complemented by Egger's regression test. An asymmetrical funnel plot or a significant Egger's test will suggest possible publication bias. All statistical analyses will be performed using R software and STATA 18.0 for more advanced statistical procedures, as required. A $p < 0.05$ will be considered statistically significant. A detailed presentation of the synthesis results will be provided through tables, forest plots, and other relevant graphical representations.

This proposed living systematic review and meta-analysis will be updated every 3–5 years as more data emerges.

## Discussion

This will be the first comprehensive review to estimate the burden of CPA along the continuum of PTB care. It is now widely accepted that CPA may be confused with, or a co-infection of PTB, or may manifest itself as a complication following anti-TB therapy [17]. However, very few prospective cohort studies have been conducted to explore evolution (such as progression and or spontaneous resolution of CPA) and emergence of new CPA cases during and after PTB therapy [5, 10, 25, 26]. Furthermore, due to lack of evidence, there is no consensus on the timing of when PTB associated CPA should be screened.

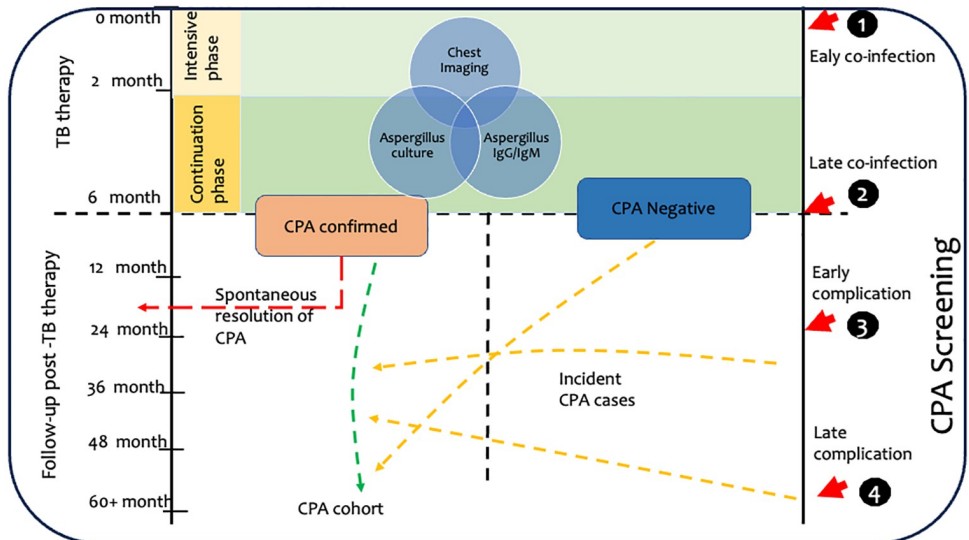

**Fig 1. Proposed screening schedules and clinical cohort establishment for chronic pulmonary aspergillosis associated with pulmonary tuberculosis.**

Denning and Ray [17] recently proposed four periods for the estimation of CPA burden: 1) early (2–6 months during anti-tuberculous therapy); 2) late (7–12 months after starting ATT); 3) 2–5 years; and 4) more than 5 years after TB diagnosis. This classification is based on international TB statistics, which typically report 12-month mortality rates, and the recognition that a diagnosis of CPA is unlikely to be made within 6 months but may become apparent during the 7–12 months following diagnosis and the initiation of anti-tuberculosis therapy. This is primarily due to the fact that CPA is a clinically subtle disease, especially in its early stages.

In settings with high TB and HIV burden with wide array of opportunistic infections, with symptoms overlapping those of CPA [27], serological investigations such as *Aspergillus*-specific IgG, with a very high sensitivity and specificity is very useful in ruling in or out CPA [28, 29]. We propose a simple screening schedule for the diagnosis of CPA associated with PTB, see Fig 1.

## Study limitations and strengths

We anticipate several limitations with this proposed systematic review and meta-analysis. Firstly, there will be variable timing for the diagnosis of CPA following the diagnosis and treatment initiation for PTB, which may reduce the accuracy of pooling prevalence data. Secondly, we expect significant heterogeneity in the pooled prevalence due to the diversity of the PTB population, timing of CPA diagnosis, geographic location, and varying study sizes in the included studies. Thirdly, conducting sensitivity analyses and sub-analyses may be challenging in certain planned analyses due to a limited number of studies or a small number of included studies. Nevertheless, we will exclusively consider studies that enrolled more than 10 patients in PTB populations or clearly indicated the involvement of PTB patients. Additionally, the diagnosis of CPA must be clearly stated and consistent with established international criteria.

## Conclusions

In summary, this intended living systematic review aims to consolidate the existing evidence concerning the burden of CPA at various stages of PTB care: at the time of PTB diagnosis,

during PTB treatment, and post-PTB treatment and will be updated regularly to inform research and practice. The outcomes of this study hold the promise of informing healthcare policies regarding CPA screening, improving clinical decision-making, and stimulating additional research to unravel the complex relationship between PTB and CPA. By illuminating the prevailing burden of CPA throughout the spectrum of PTB care, our goal is to make a meaningful contribution to the enhancement of patient care, disease management, and global health outcomes.

## Supporting information

**S1 Checklist. Reporting checklist for protocol of a systematic review and meta-analysis.** (DOCX)

## Author Contributions

**Conceptualization:** Felix Bongomin, David W. Denning.

**Data curation:** Felix Bongomin, Winnie Kibone.

**Formal analysis:** Felix Bongomin, Ronald Olum, Martha Namusobya.

**Funding acquisition:** Felix Bongomin, David W. Denning.

**Investigation:** Felix Bongomin, Winnie Kibone, Martha Namusobya, Norman van Rhijn, David W. Denning.

**Methodology:** Felix Bongomin, Ronald Olum, Winnie Kibone, Martha Namusobya, Norman van Rhijn, David W. Denning.

**Project administration:** Felix Bongomin.

**Resources:** Felix Bongomin.

**Software:** Felix Bongomin.

**Supervision:** Felix Bongomin, Norman van Rhijn, David W. Denning.

**Validation:** Felix Bongomin, Ronald Olum, Winnie Kibone, Martha Namusobya, Norman van Rhijn, David W. Denning.

**Visualization:** Felix Bongomin, Ronald Olum, Martha Namusobya, Norman van Rhijn, David W. Denning.

**Writing – original draft:** Felix Bongomin, Ronald Olum, Winnie Kibone, Martha Namusobya, Norman van Rhijn, David W. Denning.

**Writing – review & editing:** Felix Bongomin, Ronald Olum, Winnie Kibone, Martha Namusobya, Norman van Rhijn, David W. Denning.

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
