## [Decision Letter · Decision Letter 0]

27 Oct 2023

PONE-D-23-30285Prevalence of chronic pulmonary aspergillosis along the continuum of pulmonary tuberculosis care: a protocol for a living systematic review and meta-analysisPLOS ONE

Dear Dr. Bongomin,

Thank you for submitting your manuscript to PLOS ONE. After careful consideration, we feel that it has merit but does not fully meet PLOS ONE’s publication criteria as it currently stands. Therefore, we invite you to submit a revised version of the manuscript that addresses the points raised during the review process.

We look forward to receiving your revised manuscript.

Kind regards,

Aleksandra Barac

Academic Editor

PLOS ONE

Journal Requirements:

Reviewers' comments:

Reviewer's Responses to Questions

**Comments to the Author**

1. Does the manuscript provide a valid rationale for the proposed study, with clearly identified and justified research questions?

Reviewer #1: Yes

Reviewer #2: Yes

2. Is the protocol technically sound and planned in a manner that will lead to a meaningful outcome and allow testing the stated hypotheses?

Reviewer #1: Yes

Reviewer #2: Yes

3. Is the methodology feasible and described in sufficient detail to allow the work to be replicable?

Reviewer #1: Yes

Reviewer #2: Yes

4. Have the authors described where all data underlying the findings will be made available when the study is complete?

Reviewer #1: Yes

Reviewer #2: Yes

5. Is the manuscript presented in an intelligible fashion and written in standard English?

Reviewer #1: Yes

Reviewer #2: Yes

6. Review Comments to the Author

You may also provide optional suggestions and comments to authors that they might find helpful in planning their study.

Reviewer #1: The proposed article introduces a living systematic review protocol focused on determining the prevalence of Chronic Pulmonary Aspergillosis (CPA) within the continuum of Pulmonary Tuberculosis (TB) care. The protocol is designed to address an important knowledge gap in this area and improve our understanding of the coexistence of CPA and TB in various stages of care.

I find that the research questions are precise, the methodology seems robust and well-structured, and a comprehensive search strategy gives me confidence in the review's ability to provide valuable insights for both researchers and clinicians. Bias and data analysis is clearly outlined and seems robust.

I recommend the publication of this protocol, as it promises to be a good resource that will ultimately improve the care of patients dealing with this condition.

Reviewer #2: It is an excellent proposal considering that there is a complex interplay between CPA and PTB/Post TB patients.there could be a lot of heterogeneity in the results due to varying sizes of studies, ethnicity,geographic location, timing,population diversity which already has been stated by the authors.For post TB patients, do the authors wish to include those studies only which have done a CT scan and Gene X pert for all the patients?What about immunocompromised patients?ALL in all ,it is a wonderful study and I wish them good luck.

7. PLOS authors have the option to publish the peer review history of their article (what does this mean?). If published, this will include your full peer review and any attached files.

Reviewer #1: No

Reviewer #2: No

---

## [Author Response · Author response to Decision Letter 0]

1 Nov 2023

Response to Reviewers

PONE-D-23-30285

Prevalence of chronic pulmonary aspergillosis along the continuum of pulmonary tuberculosis care: a protocol for a living systematic review and meta-analysis

PLOS ONE

Dear Dr Aleksandra Barac

Academic Editor, PLoS ONE

Thank you very much for the shift review of this protocol.

We have addressed all the reviewers’ comments and hope this protocol is now acceptable. 

Journal Requirements:

Data Availability statement,

Data Availability: All relevant data are within the article and its Supporting information files.

Reviewer #1: The proposed article introduces a living systematic review protocol focused on determining the prevalence of Chronic Pulmonary Aspergillosis (CPA) within the continuum of Pulmonary Tuberculosis (TB) care. The protocol is designed to address an important knowledge gap in this area and improve our understanding of the coexistence of CPA and TB in various stages of care.

I find that the research questions are precise, the methodology seems robust and well-structured, and a comprehensive search strategy gives me confidence in the review's ability to provide valuable insights for both researchers and clinicians. Bias and data analysis is clearly outlined and seems robust.

I recommend the publication of this protocol, as it promises to be a good resource that will ultimately improve the care of patients dealing with this condition.

Authors’ response: Thank you very much for commending our protocol. 

Reviewer #2: It is an excellent proposal considering that there is a complex interplay between CPA and PTB/Post TB patients.there could be a lot of heterogeneity in the results due to varying sizes of studies, ethnicity, geographic location, timing,population diversity which already has been stated by the authors. For post TB patients, do the authors wish to include those studies only which have done a CT scan and Gene X pert for all the patients?What about immunocompromised patients?ALL in all ,it is a wonderful study and I wish them good luck.

Authors’ response: Thank you very much. We will include both immuncompromised and immunocompetent individuals in whom all the relevant investigations consistent with the diagnostic work up for CPA have been document. We surely appreciate the heterogeneity and complexity of CPA.

Kind regards

Dr Felix Bongomin

---

## [Editor Report · Decision Letter 1]

6 Nov 2023

Prevalence of chronic pulmonary aspergillosis along the continuum of pulmonary tuberculosis care: a protocol for a living systematic review and meta-analysis

PONE-D-23-30285R1

Dear Dr. Bongomin,

We’re pleased to inform you that your manuscript has been judged scientifically suitable for publication and will be formally accepted for publication once it meets all outstanding technical requirements.

Kind regards,

Aleksandra Barac

Academic Editor

PLOS ONE

---

## [Editor Report · Acceptance letter]

6 Dec 2023

PONE-D-23-30285R1 

Prevalence of chronic pulmonary aspergillosis along the continuum of pulmonary tuberculosis care: a protocol for a living systematic review and meta-analysis 

Dear Dr. Bongomin:

I'm pleased to inform you that your manuscript has been deemed suitable for publication in PLOS ONE. Congratulations! Your manuscript is now with our production department. 

Kind regards, 

on behalf of

Dr. Aleksandra Barac 

Academic Editor

PLOS ONE